# A simple, single-tube overlapping amplicon-targeted Illumina sequencing assay

**Jason D. Limberis**[ID]¹*, **Alina Nalyvayko**¹, **Joel D. Ernst**¹, **John Z. Metcalfe**²

**1** Division of Experimental Medicine, University of California, San Francisco, San Francisco, California, United States of America, **2** Division of Pulmonary and Critical Care Medicine, Zuckerberg San Francisco General Hospital and Trauma Centre, University of California, San Francisco, San Francisco, California, United States of America

* jason.limberis@ucsf.edu

## Abstract

Targeted amplicon sequencing to identify pathogens, resistance-conferring mutations, and strain types is an important tool in diagnosing and treating infections. However, due to the short read limitations of Illumina sequencing, many applications require the splitting of limited clinical samples between two reactions. Here, we outline hairpin Illumina single-tube sequencing PCR (*hiss*PCR) which allows for the generation of overlapping amplicons containing Illumina indexes and adapters in a single tube, effectively extending the Illumina read length while maintaining reagent and sample input requirements.

## Introduction

Targeted amplicon sequencing enables the identification of resistance-conferring mutations and strain typing of important human pathogens. However, due to the short read limitations of Illumina sequencing–a maximum of 600 base pairs (bp) when 300bp paired-end sequencing is done–regions larger than 600bp require overlapping amplicons that necessitate splitting a sample between two reactions [1]. However, clinical samples are often limited, and nucleic acid concentrations are difficult to quantify, increasing the chance of sequencing failure. For example, DNA extraction of *Mycobacterium tuberculosis*, the etiological agent of tuberculosis, is most commonly done from patient sputum. This results in the co-extraction of human DNA including oral and respiratory flora, thus complicating the quantification of *M. tuberculosis* DNA in a sample. Furthermore, with as few as 5,000 bacteria/ml in the sputum of a culture positive patient [2], there is limited material to work with. Additionally, standard library preparation is complex [3] and requires numerous enzymatic reactions leading to multiple vulnerabilities within targeted amplicon sequencing workflows and complicating point-of-care or near point-of-care applications.

Here, we outline a hairpin Illumina single-tube sequencing PCR (*hiss*PCR), a method that allows for the generation of sequenceable, overlapping amplicons in a single tube, effectively tripling Illumina read length. *hiss*PCR uses target-specific primers with universal tail sequences. In the first round of PCR, primers containing a universal tail sequence amplify a target up to 1200 nucleotides (Fig 1, **step 1**). In the second round of PCR, tailed gene-specific

**Funding:** JZM, 1R01AI153213, National Institute of Allergy and Infectious Diseases (NIAID), https://www.niaid.nih.gov/, The funders did not and will not have a role in study design, data collection and analysis, decision to publish, or preparation of the manuscript.

**Competing interests:** The authors have declared that no competing interests exist.

**Fig 1. Hairpin Illumina single-tube sequencing PCR (*hiss*PCR).** In step 1, primers containing a universal tail sequence amplify a target up to 1200 nucleotides flanking each gene of interest. In step 2, tailed gene-specific primers target the amplicon generated in step 1, and, together with the universal tailed primers that contain the index and adapter sequence, generate two overlapping amplicons of sequenceable length while the gene-specific primers (Forward-1 [F1] and Reverse-2 [R2], each containing the same tailed sequence), form a hairpin during amplification, thus nullifying the formation of the 100 base pair overlap amplicon and allowing amplification of the intended ~600 base pair amplicons.

primers target the amplicon generated in step 1. Together with the universal tailed primers containing the index and adapter sequence, generate two overlapping amplicons of sequence-able length (Fig 1, **step 2**). In a traditional PCR, the overlapping primers (F2 and R2) would amplify not only the original template but also subsequent amplicons, overtaking the reaction with amplicons ~100 nucleotides in length. In *hiss*PCR however, the gene-specific primers (F2 and R2) contain the same tailed sequence. These tails form a hairpin during amplification (Fig 1, **step 2**), nullifying the formation of the 100 base pair overlap amplicon and allowing amplification of the intended ~600 base pair amplicons. Paired-end sequencing (Fig 1, **step 3**) then generates the full 1200 base pair sequence. Several of these primers can be designed to overlap, covering even larger areas.

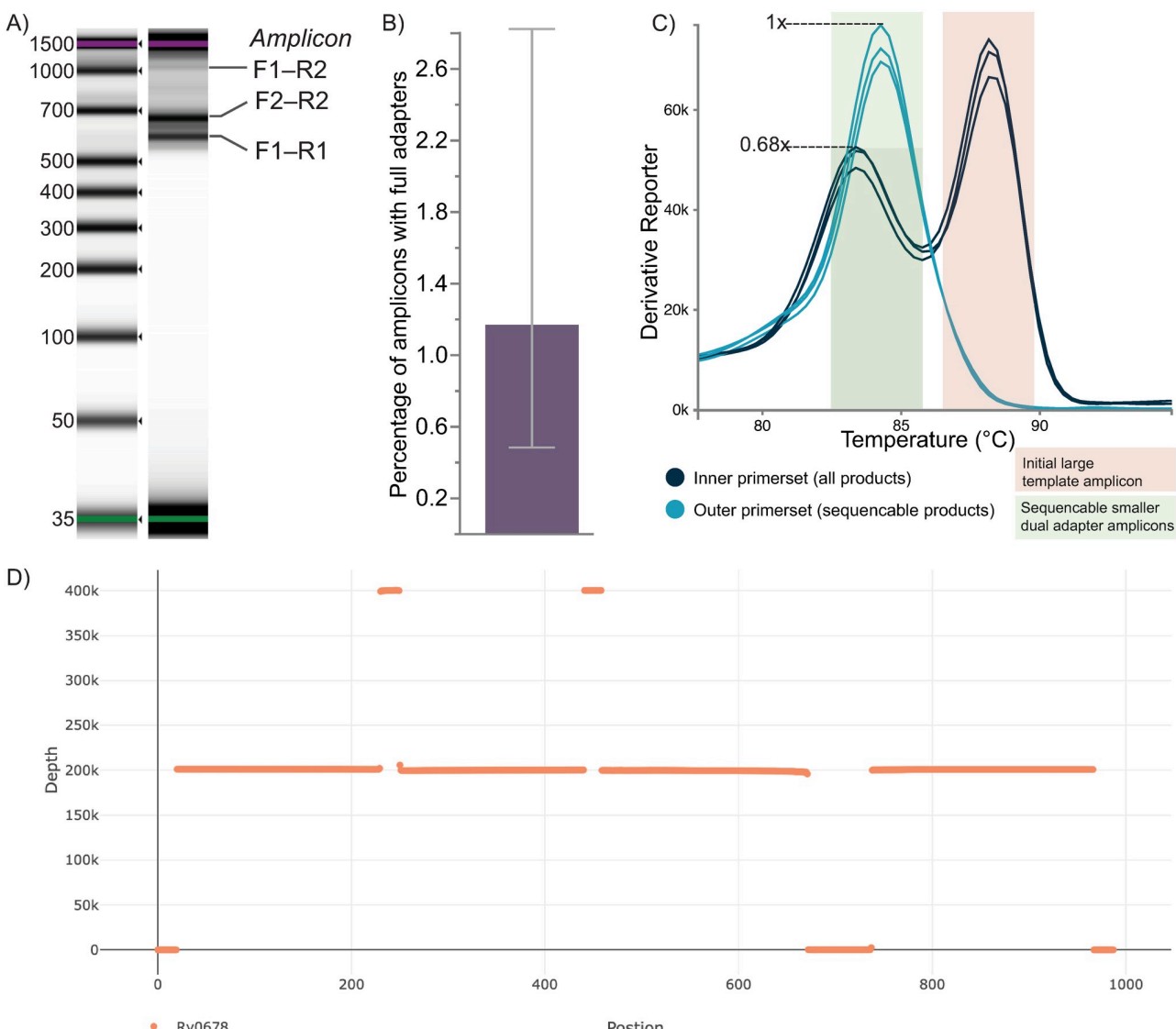

**Fig 2. Expected results.** A) *hiss*PCR generated amplicons using two forward and two reverse overlapping primers in a single tube. Amplicons contain either no Illumina adapter, one Illumina adapter, or two Illumina adapters per amplicon. B) The relative proportion of amplicons in the sample containing both Illumina adapters. C) Melt curves showing the expected amplicons for each primer set and the ratio of amplicons containing both Illumina adapters.

## Materials and methods

The protocol described in this peer-reviewed article is published on protocols.io, DOI 10.17504/protocols.io.q26g7yw79gwz/v1, and is included for printing as S1 File with this article.

## Expected results

There will be several amplicons for each target (Fig 2). The amplicons will include the large amplicon, and the two small amplicons, with no Illumina adapters, an Illumina adapter on one end of the amplicon, or Illumina adapters on both ends of the amplicon (the sequenceable amplicons). The large amplicons are not sequenceable as they have the same adapter at both ends, even if it is generated in the second PCR using the Illumina adapter primers. The qPCR ratio of the Illumina inner and outer primer sets will provide the proportion of DNA in the sample with Illumina adapters on both ends of the amplicon (the sequenceable amplicons), while the melt curve will provide information on the proportion of each amplicon in the sample. Alternatively, amplicons can be quantified using commercial Illumina library quantification kits.

## Supporting information

**S1 File. Step-by-step protocol, also available on protocols.io, DOI 10.17504/protocols.io.q26g7yw79gwz/v1.**
(PDF)

**S2 File. Raw output from D1000 Agilent TapeStation of hissPCR generated amplicons using two forward and two reverse overlapping primers in a single tube as shown in Fig 2A.**
(CSV)

**S3 File.**
(PDF)

**S1 Raw images.**
(PDF)

## Author Contributions

**Formal analysis:** Jason D. Limberis.

**Funding acquisition:** John Z. Metcalfe.

**Investigation:** Jason D. Limberis, Alina Nalyvayko.

**Methodology:** Jason D. Limberis.

**Project administration:** John Z. Metcalfe.

**Supervision:** John Z. Metcalfe.

**Visualization:** Jason D. Limberis.

**Writing – original draft:** Jason D. Limberis, John Z. Metcalfe.

**Writing – review & editing:** Jason D. Limberis, Alina Nalyvayko, Joel D. Ernst, John Z. Metcalfe.

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
