## [Decision Letter · Decision Letter 0]

23 Mar 2023

PONE-D-23-05939A simple, single-tube overlapping amplicon-targeted Illumina sequencing assay.PLOS ONE

Dear Dr. Limberis,

Thank you for submitting your manuscript to PLOS ONE. After careful consideration, we feel that it has merit but does not fully meet PLOS ONE’s publication criteria as it currently stands. Therefore, we invite you to submit a revised version of the manuscript that addresses the points raised during the review process.

We look forward to receiving your revised manuscript.

Kind regards,

Padmapriya P Banada, PhD

Academic Editor

PLOS ONE

Journal Requirements:

3. We note you have not yet provided a protocols.io PDF version of your protocol and/or a protocols.io DOI. When you submit your revision, please provide a PDF version of your protocol as generated by protocols.io (the file will have the protocols.io logo in the upper right corner of the first page) as a Supporting Information file. The filename should be S1_file.pdf, and you should enter “S1 File” into the Description field. Any additional protocols should be numbered S2, S3, and so on. Please also follow the instructions for Supporting Information captions [https://journals.plos.org/plosone/s/supporting-information#loc-captions]. The title in the caption should read: “Step-by-step protocol, also available on protocols.io.”

Please assign your protocol a protocols.io DOI, if you have not already done so, and include the following line in the Materials and Methods section of your manuscript: “The protocol described in this peer-reviewed article is published on protocols.io (https://dx.doi.org/10.17504/protocols.io.[...]) and is included for printing purposes as S1 File.” You should also supply the DOI in the Protocols.io DOI field of the submission form when you submit your revision.

If you have not yet uploaded your protocol to protocols.io, you are invited to use the platform’s protocol entry service [https://www.protocols.io/we-enter-protocols] for doing so, at no charge. Through this service, the team at protocols.io will enter your protocol for you and format it in a way that takes advantage of the platform’s features. When submitting your protocol to the protocol entry service please include the customer code PLOS2022 in the Note field and indicate that your protocol is associated with a PLOS ONE Lab Protocol Submission. You should also include the title and manuscript number of your PLOS ONE submission.

Additional Editor Comments (if provided):

Thank you for submitting your protocol and I hope the comments from the reviewers would help improve the methodology and increased success of the proposal. In general, I agree with the reviewers that the protocol limitations and potential solutions should be clearly mentioned. Although reviewer 1 has major concerns on the amplicon size and illumina limitations, please expand on this clearly in your introduction.

Reviewers' comments:

Reviewer's Responses to Questions

**Comments to the Author**

1. Does the manuscript report a protocol which is of utility to the research community and adds value to the published literature?

Reviewer #1: Yes

Reviewer #2: Yes

Reviewer #3: Yes

2. Has the protocol been described in sufficient detail?

To answer this question, please click the link to protocols.io in the Materials and Methods section of the manuscript (if a link has been provided) or consult the step-by-step protocol in the Supporting Information files.

The step-by-step protocol should contain sufficient detail for another researcher to be able to reproduce all experiments and analyses.

Reviewer #1: No

Reviewer #2: Partly

Reviewer #3: Yes

3. Does the protocol describe a validated method?

Reviewer #1: No

Reviewer #2: No

Reviewer #3: No

4. If the manuscript contains new data, have the authors made this data fully available?

Reviewer #1: No

Reviewer #2: N/A

Reviewer #3: No

**5. Is the article presented in an intelligible fashion and written in standard English?**

Reviewer #1: Yes

Reviewer #2: Yes

Reviewer #3: Yes

6. Review Comments to the Author

Reviewer #1: The manuscript A simple, single-tube overlapping amplicon-targeted Illumina sequencing assay, suggests novel approach for generating longer amplicons using illumina sequencing. Their approach uses a hairpin Illumina single-tube sequencing PCR (hissPCR), a method that allows for the generation of sequenceable, overlapping amplicons in a single tube, effectively tripling Illumina read length.

Although their approach aims to address a critical issue in the field, they do not provide any proof that their approach works. They mention in their intro that with 300bp paired end sequencing, they can sequence an amplicon that is 600bp long. However, paired end sequencing is designed to produce high quality coverage for the region being sequenced (300bp) not 600bp. Furthermore, 300bp paired end sequencing is limited to certain sequencing instruments which should be mentioned in their intro and methods.

Furthermore, illumina sequencing was designed to be accurate for short reads and the authors do not explain how they plan to overcome issues in accuracy with longer reads or how they plan to even sequence longer reads (single read vs paired reads, specific instruments, library prep, etc). They also do not provide any information on expected read coverage and how those reads could be potentially used for downstream analysis particularly for TB clinical assays.

Reviewer #2: Dear authors, I have following suggestions to improve the manuscript:

1) Figure 1: Step 2 please align the R1 primers in the F2R1 amplicon with the R1 primers in the F1R1 amplicon. It looks misleading as it is.

2) Figure 1: Step 3: please correct spelling of "generation".

3) Methods and material: Please describe in detail target identification and primer design for a single-plex reaction sequenceable overlapping amplicons, to ensure the hairpin structure of short 100bp amplicon is retained.

4) Expected results:

a. Please describe in detail any short comings of this method, essential precautions partaken to ensure minimum contamination and best output, including but not limited to QC.

b. Please include comments on the bioinformatics pipeline and computational methods used for clean up of the data.

c. Finally, please also include comments on whether single -plex reaction has impact on improved efficiency, coverage and cost of the sequencing.

Reviewer #3: Manuscript present an new PCR assay to amply longer genomic intervals in microbial genome using a nested PCR strategy. The approach is not entirely novel but protocol developed by authors does makes sense and is described appropriately. I have following comments and questions for authors:

1. Given that amplified product contains fragments of multiple sizes, precise quantification of library before sequencing may be challenging and would need accurate nanomole calculation.

2. Authors have not shown how efficiently they were able to clean the longer amplicons using bead ratio? A tapestation image of library after the bead clean up would be helpful. A band of high molecular weight is evident in the amplicon lane.

3. Accurate library quantification before loading the flow cell is crucial for optimal cluster generation on target. It may cause under or over clustering and failure of sequencing run!

4. Would assay perform similarly in genomes with different GC ratio and intervals with more repetitive sequences or structural variants? Any thoughts?

5. Direct whole genome sequencing using long-read sequencing method is feasible now. What authors think of those options to address this question instead of classical nested PCR approach.

6. Would time and reagent costs in presented protocol vs whole genome sequencing approach, would be significantly different? Any thoughts

7. Manuscript does not provide any data generated using this protocol. How did pathogen identification, taxonomic classification and mutation detection analysis go ? It is important to validate the intended application of this protocol?

Overall, manuscript does present a new protocol to amplify targeted regions in the pathogen genome. If authors agree to address raised points, I would recommend publication of this manuscript.

7. PLOS authors have the option to publish the peer review history of their article (what does this mean?). If published, this will include your full peer review and any attached files.

Reviewer #1: No

Reviewer #2: No

Reviewer #3: **Yes: **Prithvi Raj, Ph.D. Microbiome Research Lab, UT Southwestern Medical Center, Dallas, Texas, USA

---

## [Author Response · Author response to Decision Letter 0]

16 May 2023

PONE-D-23-05939

A simple, single-tube overlapping amplicon-targeted Illumina sequencing assay.

PLOS ONE

Additional Editor Comments (if provided):

Thank you for submitting your protocol and I hope the comments from the reviewers would help improve the methodology and increased success of the proposal. In general, I agree with the reviewers that the protocol limitations and potential solutions should be clearly mentioned. Although reviewer 1 has major concerns on the amplicon size and illumina limitations, please expand on this clearly in your introduction.

Thank you for the opportunity to submit a revised protocol, and we appreciate the reviewers' comments and suggestions. We have carefully considered each comment and made the necessary revisions to strengthen the manuscript. Reviewer 1 raised concerns about the amplicon size and Illumina limitations. We have expanded the introduction to provide a clear explanation and have included supporting sequencing data to demonstrate the success of the method. Reviewer 2 suggested adding more detail on the sample analysis process and Reviewer 3 recommended including accurate library quantification all of which are now included in the manuscript. We have responded to the specific concerns below (in blue text) and made the relevant changes to the manuscript and protocol. These revisions have significantly improved the quality of the manuscript, and we hope that you and the reviewers find them satisfactory. Thank you for your consideration, and we look forward to hearing from you.

Reviewers' comments:

Reviewer's Responses to Questions

Comments to the Author

1. Does the manuscript report a protocol which is of utility to the research community and adds value to the published literature?

Reviewer #1: Yes

Reviewer #2: Yes

Reviewer #3: Yes

2. Has the protocol been described in sufficient detail?

To answer this question, please click the link to protocols.io in the Materials and Methods section of the manuscript (if a link has been provided) or consult the step-by-step protocol in the Supporting Information files.

The step-by-step protocol should contain sufficient detail for another researcher to be able to reproduce all experiments and analyses.

Reviewer #1: No

Reviewer #2: Partly

Reviewer #3: Yes

3. Does the protocol describe a validated method?

Reviewer #1: No

Reviewer #2: No

Reviewer #3: No

4. If the manuscript contains new data, have the authors made this data fully available?

Reviewer #1: No

Reviewer #2: N/A

Reviewer #3: No

5. Is the article presented in an intelligible fashion and written in standard English?

Reviewer #1: Yes

Reviewer #2: Yes

Reviewer #3: Yes

6. Review Comments to the Author

Reviewer #1: The manuscript A simple, single-tube overlapping amplicon-targeted Illumina sequencing assay, suggests novel approach for generating longer amplicons using illumina sequencing. Their approach uses a hairpin Illumina single-tube sequencing PCR (hissPCR), a method that allows for the generation of sequenceable, overlapping amplicons in a single tube, effectively tripling Illumina read length.

Although their approach aims to address a critical issue in the field, they do not provide any proof that their approach works. They mention in their intro that with 300bp paired end sequencing, they can sequence an amplicon that is 600bp long. However, paired end sequencing is designed to produce high quality coverage for the region being sequenced (300bp) not 600bp. Furthermore, 300bp paired end sequencing is limited to certain sequencing instruments which should be mentioned in their intro and methods.

Thank you for your concerns. We have added supporting sequencing data to demonstrate the success of the method. The output of the alignments is now in Figure 2. Regarding the sequencing length, when performing pair-end sequencing, each read in the pair is sequenced the number of cycles programmed (i.e., for 250bp paired end sequencing, a 500bp region of the individual DNA, 250bp from each end). Thus, if an amplicon of 500bp is sequenced, the entire 500bp will be reported; when 300bp paired end sequencing is used, the maximum coverage of a single amplicon is 600bp.

Line 19: Targeted amplicon sequencing enables the identification of resistance-conferring mutations and strain typing of important human pathogens. However, due to the short read limitations of Illumina sequencing – a maximum of 600 base pairs (bp) when 300bp paired-end sequencing is done – regions larger than 600bp require overlapping amplicons that necessitate splitting a sample between two reactions [1].

Line 45: Three hundred bp paired-end sequencing (Figure 1, step 3) then generates the full 1200 base pair sequence (an amplicon of up to four times the sequencing length can be sequenced per reaction, e.g., 4 x 250bp paired-end sequencing = 1000bp). Several of these primers can be designed to overlap, covering even larger areas.

Line 90: D) An output from the hissPCR analysis pipeline showing the expected read coverage over the target area using 250bp Illumina sequencing.

Furthermore, illumina sequencing was designed to be accurate for short reads and the authors do not explain how they plan to overcome issues in accuracy with longer reads or how they plan to even sequence longer reads (single read vs paired reads, specific instruments, library prep, etc). 

Illumina sequencing is highly accurate but has limited read length (common kits allow for 50, 75, 150, 250, and 300bp sequencing reactions). However, the length of the DNA sequenced does not adversely affect the process or quality (https://knowledge.illumina.com/library-preparation/general/library-preparation-general-reference_material-list/000003874), and several Illumina PCR amplicon-based kits sequence short (275bp) PCR products (AmpliSeq product line, https://www.illumina.com/products/by-brand/ampliseq.html). It is also possible, using paired end sequencing, to sequence the same portion of DNA from both directions (reads) when the fragment is smaller than the read length (Figure below). This is used in some cancer assays to increase confidence in the base calls and allow for the detection of minor heterogeneous populations as is common in cancer samples.

They also do not provide any information on expected read coverage and how those reads could be potentially used for downstream analysis particularly for TB clinical assays.

This is determined by the end-user; however, we have added some guidelines to the protocol (“Depth and Pooling Calculations”) detailing how to accurately calculate the library for optimal flow cell loading and sequencing.

Reviewer #2: Dear authors, I have following suggestions to improve the manuscript:

1) Figure 1: Step 2 please align the R1 primers in the F2R1 amplicon with the R1 primers in the F1R1 amplicon. It looks misleading as it is.

2) Figure 1: Step 3: please correct spelling of "generation".

3) Methods and material: Please describe in detail target identification and primer design for a single-plex reaction sequenceable overlapping amplicons, to ensure the hairpin structure of short 100bp amplicon is retained.

Thank you for your comments, we have edited Figure 1 for clarity and incorporated your suggestions. We have also added a section to the protocol (“Primer design”) that details the primer design process and have published a pipeline on GitHub to assist users with their design (https://github.com/SemiQuant/hissPCR).

4) Expected results:

a. Please describe in detail any short comings of this method, essential precautions partaken to ensure minimum contamination and best output, including but not limited to QC.

We have added the shortcomings to the manuscript: the main limitation is the possibility of uneven amplicon ratios in the sample, which we specifically address in the manuscript. 

Line 70: “While hissPCR should not adversely affect Illumina cluster generation and sequencing, its is important to accurately quantify the sequenceable material to load the appropriate amount on the flow cell; however, since not all the amplicons will be present at the same ratios, we recommend aiming for a 1.2-1.5x higher than the desired coverage (this will be especially important for multiplexed hissPCR reactions).”

b. Please include comments on the bioinformatics pipeline and computational methods used for clean up of the data.

We have added a section to the protocol (“Data Analysis”) that details the methodology to analyze the Illumina sequencing data produces. We have also published a pipeline on GitHub to assist users with their analyses.

Line 73: The sequence data can be analyzed using most amplicon processing pipelines; however, the primer sequences must be trimmed from the reads to eliminate their effect on mutation identification. Our analysis pipeline utilizes the `samtools ampliconclip` command to remove these.

c. Finally, please also include comments on whether single -plex reaction has impact on improved efficiency, coverage and cost of the sequencing.

We have included a comment on how hissPCR does not affect the efficiency of sequencing. However, coverage and cost of sequencing are subject to numerous factors, which will differ depending on how the user utilizes the method.

Reviewer #3: Manuscript present an new PCR assay to amply longer genomic intervals in microbial genome using a nested PCR strategy. The approach is not entirely novel but protocol developed by authors does makes sense and is described appropriately. I have following comments and questions for authors:

1. Given that amplified product contains fragments of multiple sizes, precise quantification of library before sequencing may be challenging and would need accurate nanomole calculation.

2. Authors have not shown how efficiently they were able to clean the longer amplicons using bead ratio? A tapestation image of library after the bead clean up would be helpful. A band of high molecular weight is evident in the amplicon lane.

3. Accurate library quantification before loading the flow cell is crucial for optimal cluster generation on target. It may cause under or over clustering and failure of sequencing run!

Responses 1-3: Thank you for these comments and suggestions. We have added in a section to the protocol (“Depth and Pooling Calculations”) detailing how to accurately calculate the library for optimal flow cell loading and sequencing. The figure in the protocol shows the amplicons post bead cleanup, while these do still contain non-sequenceable material, they will not affect the Illumina cluster generation as they fail to bind to the flowcell.

4. Would assay perform similarly in genomes with different GC ratio and intervals with more repetitive sequences or structural variants? Any thoughts?

We expect hissPCR to perform as a regular PCR if the annealing temperature is >55C, regardless of the GC content. The hairpin is formed predominantly by the tail sequences which are constant, so structural variants should also perform like regular end-point PCR.

5. Direct whole genome sequencing using long-read sequencing method is feasible now. What authors think of those options to address this question instead of classical nested PCR approach.

While whole genome sequencing using long-read sequencing is feasible, it requires a large input of DNA which is not present in all clinical samples and needs more sequencing reads than targeted sequencing. Furthermore, in many clinical samples (e.g., sputa), the majority of the DNA is from the host rather than the pathogen of interest, rendering direct whole genome sequencing difficult. 

6. Would time and reagent costs in presented protocol vs whole genome sequencing approach, would be significantly different? Any thoughts

The reagent cost would be less than regular, split sample PCR, and on the order of what whole genome sequencing approaches. However, this difference will depend on numerous factors chosen by the user, including sequencing depth, genome and target size, and the number of samples available for pooling. 

7. Manuscript does not provide any data generated using this protocol. How did pathogen identification, taxonomic classification and mutation detection analysis go? It is important to validate the intended application of this protocol?

We have included sequencing results now in Figure 2 (panel D) and found base calls the same as end-point PCR.

Overall, manuscript does present a new protocol to amplify targeted regions in the pathogen genome. If authors agree to address raised points, I would recommend publication of this manuscript.

7. PLOS authors have the option to publish the peer review history of their article (what does this mean?). If published, this will include your full peer review and any attached files.

Do you want your identity to be public for this peer review? For information about this choice, including consent withdrawal, please see our Privacy Policy.

Reviewer #1: No

Reviewer #2: No

Reviewer #3: Yes: Prithvi Raj, Ph.D. Microbiome Research Lab, UT Southwestern Medical Center, Dallas, Texas, USA

---

## [Editor Report · Decision Letter 1]

2 Jul 2023

A simple, single-tube overlapping amplicon-targeted Illumina sequencing assay.

PONE-D-23-05939R1

Dear Dr. Limberis,

We’re pleased to inform you that your manuscript has been judged scientifically suitable for publication and will be formally accepted for publication once it meets all outstanding technical requirements.

Kind regards,

Padmapriya P Banada, PhD

Academic Editor

PLOS ONE

Additional Editor Comments (optional):

Thank you for resubmitting your manuscript and considering reviewers comments positively. I do see that the methodology described is more accurate, clear and understandable. I accept that rebuttal provided for all the reviewers comments.
---

## [Editor Report · Acceptance letter]

5 Sep 2023

PONE-D-23-05939R1 

A simple, single-tube overlapping amplicon-targeted Illumina sequencing assay. 

Dear Dr. Limberis:

I'm pleased to inform you that your manuscript has been deemed suitable for publication in PLOS ONE. Congratulations! Your manuscript is now with our production department. 

Kind regards, 

on behalf of

Dr. Padmapriya P Banada 

Academic Editor

PLOS ONE